# Cellular and Molecular Interactions in CNS Injury: The Role of Immune Cells and Inflammatory Responses in Damage and Repair

**DOI:** 10.3390/cells14120918

**Published:** 2025-06-18

**Authors:** Jai Chand Patel, Meenakshi Shukla, Manish Shukla

**Affiliations:** 1Department of Genetics Cell Biology & Anatomy, University of Nebraska Medical Center, Omaha, NE 68198, USA; japatel@unmc.edu; 2Department of Neurosurgery, Penn State Milton S. Hershey Medical Center, Hershey, PA 17033, USA; mshukla2@pennstatehealth.psu.edu

**Keywords:** CNS injury, neurodegenerative diseases, inflammation, microglia, excitotoxicity, neurogenesis

## Abstract

The central nervous system (CNS) is highly susceptible to damage due to its limited ability to regenerate. Injuries to the CNS, whether from trauma, ischemia, or neurodegenerative diseases, disrupt both cellular and vascular structures, leading to immediate (primary) and subsequent (secondary) damage. Primary damage involves the physical disruption of cells and blood vessels, weakening the blood–brain barrier (BBB) and triggering excitotoxicity and calcium overload. Secondary damage develops over hours to days and is marked by ionic imbalance, mitochondrial dysfunction, oxidative stress, and chronic inflammation, which further aggravates tissue damage. Inflammation plays a dual role: acute inflammation helps in repair, while chronic inflammation accelerates neurodegeneration. Microglia and astrocytes play key roles in this inflammatory response, with M1-like microglia promoting pro-inflammatory responses and M2-like microglia supporting anti-inflammatory and repair processes. Neurodegenerative diseases are characterized by the accumulation of misfolded proteins such as Tau, amyloid-beta, TDP-43, and α-synuclein, which impair cellular function and lead to neuronal loss. Neurodegenerative diseases are characterized by the accumulation of misfolded proteins and influenced by genetic risk factors (e.g., *APOE4, TARDBP*). Despite the CNS’s limited regenerative abilities, processes like synaptogenesis, neurogenesis, axonal regeneration, and remyelination offer potential for recovery. Therapeutic approaches aim to target inflammatory pathways, enhance repair mechanisms, and develop neuroprotective treatments to counter excitotoxicity, oxidative stress, and apoptosis. Advances in stem cell therapy, gene therapy, and personalized medicine hold promise for improving outcomes. Future research should focus on combining strategies, utilizing advanced technologies, and conducting translational studies to bridge the gap between preclinical research and clinical application. By better understanding and leveraging the complex processes of CNS injury and repair, researchers hope to develop effective therapies to restore function and enhance the quality of life for individuals with CNS disorders.

## 1. Introduction

The central nervous system (CNS), consisting of the brain and spinal cord, is particularly vulnerable to injury due to its limited ability to regenerate and its intricate cellular structure [1,2]. CNS injuries can result from various causes, including traumatic events like traumatic brain injury or spinal cord injury, ischemic events such as stroke, and neurodegenerative diseases like Alzheimer’s and Parkinson’s [3]. These injuries disrupt the delicate balance of cellular and vascular integrity, leading to both immediate (primary) and delayed (secondary) damage. Primary damage occurs at the moment of injury and involves the direct mechanical disruption of cells and blood vessels, such as microvascular damage that impairs the blood–brain barrier (BBB) and cell membrane damage that triggers the release of intracellular contents and the initiation of cytotoxic processes [4,5]. Secondary damage, which develops over hours to days after the initial injury, involves a series of biochemical and cellular events that worsen tissue damage, including ionic imbalance, calcium overload, and mitochondrial dysfunction [6,7]. These disruptions contribute to neuronal loss, glial activation, and increased inflammation. The compromised BBB permits immune cells and harmful substances to infiltrate the CNS, further intensifying inflammation and neuronal injury [8,9,10].

## 2. Importance of Understanding Primary and Secondary Damage Mechanisms

Understanding the mechanisms behind primary and secondary damage in CNS injuries is vital for developing effective treatments. Primary damage, occurring immediately at the time of injury, involves the direct physical disruption of cells and blood vessels, leading to microvascular damage and breakdown of cell membranes [11]. This initial damage paves the way for secondary damage, a more complex and delayed series of biochemical and cellular events that worsen tissue injury. Secondary damage processes, such as ionic imbalance, calcium overload, and mitochondrial dysfunction, contribute to neuronal loss, ongoing inflammation, and further disruption of the blood–brain barrier [12,13,14]. Secondary damage involves biochemical events that worsen tissue injury, such as ionic imbalance, calcium overload, and mitochondrial dysfunction, which generate ROS. ROS further disrupt membranes, exacerbating BBB permeability and triggering inflammation [15]. A comprehensive understanding of these mechanisms allows researchers and clinicians to identify critical time points and molecular targets for intervention. This knowledge is crucial for developing strategies to reduce secondary damage, promote repair, and enhance functional recovery, ultimately improving the quality of life for those affected by CNS injuries.

### 2.1. Primary Damage Mechanisms

#### 2.1.1. Microvascular Damage

Microvascular damage plays a critical role in primary damage following CNS injuries, occurring immediately after the initial injury. This damage primarily impacts the BBB, a specialized structure that controls the exchange of molecules between the bloodstream and the brain [16,17,18]. The BBB consists of endothelial cells, pericytes, astrocytes, and a basement membrane, all of which collaborate to maintain CNS homeostasis [19]. When the CNS is injured, mechanical forces on the microvasculature can disrupt these components, leading to a compromised BBB [20,21] (Figure 1).

The immediate effects of microvascular damage include the leakage of blood components into brain tissue, resulting in edema. This increase in intracranial pressure can worsen tissue damage [22,23,24]. The extravasation of blood proteins like albumin into the brain can trigger inflammatory responses, further harming the surrounding neural tissue [10,25]. Moreover, the disrupted BBB permits immune cell infiltration, such as neutrophils and macrophages, which release pro-inflammatory cytokines and reactive oxygen species (ROS), contributing to secondary damage [26,27,28].

Endothelial cells lining the blood vessels are particularly susceptible to mechanical injury. When damaged, these cells may undergo apoptosis or necrosis, compromising the vascular wall. This damage can lead to hemorrhage, complicating the injury further. The breakdown of endothelial integrity also disrupts tight junctions that normally block harmful substances from passing between cells, allowing the uncontrolled entry of ions, toxins, and pathogens, which intensifies the injury [29,30,31]. Pericytes, contractile cells that encircle endothelial cells, are essential for maintaining vascular stability and regulating blood flow [32,33]. Damage to pericytes can cause the loss of vascular tone, impair blood flow, and lead to ischemia and hypoxia in the affected tissue [34,35]. Astrocytes, which contribute to the glia limitans and BBB function, also respond to vascular damage. When injured, astrocytes can become reactive, leading to glial scar formation, which hinders repair and regeneration [36,37]. The basement membrane, which provides structural support to blood vessels, can also be damaged during injury [38]. The degradation of the basement membrane by matrix metalloproteinases (MMPs) can destabilize the vascular wall and further disrupt the BBB while also impairing signaling pathways necessary for maintaining vascular integrity and promoting repair [39,40]. Mechanical forces disrupt endothelial cells, pericytes, and astrocytes, compromising BBB integrity. This allows blood components (e.g., albumin) to leak into brain tissue, triggering edema, inflammation, and immune cell infiltration (Figure 1). Pericyte damage impairs blood flow, causing ischemia; astrocyte injury promotes glial scarring [28,38].

#### 2.1.2. Mechanical Forces and Blood–Brain Barrier Disruption During Injury

While primary insults such as mechanical trauma, ischemia, or shear forces can directly rupture membranes and blood vessels, the progression to secondary damage involves a complex cascade of interrelated processes [41]. Mechanical forces, including direct shear or compression, can cause the immediate rupture of axons and vessels, while even sub-threshold biomechanical stress may destabilize the cytoskeleton and disassemble tight junctions [42]. Early metabolic failure, particularly the loss of ATP, disrupts ion gradients and impairs Na+/K+ pumps, leading to cytotoxic edema that physically distends cells and microvessels, increasing the likelihood of rupture [43]. Almost immediately, damage-associated molecular patterns (DAMPs) released from injured cells activate resident microglia and recruit peripheral immune cells, which release MMPs and ROS that degrade extracellular matrix components and tight junctions [44]. Concurrently, the loss of cerebrovascular autoregulation results in abnormal pressure dynamics such as capillary hypertension or vasospasm that further stress the compromised vasculature [45]. These mechanisms are compounded by feed-forward loops, where oxidative stress and lipid peroxidation compromise membrane integrity, allowing calcium influx that further activates proteases and ROS-generating enzymes, amplifying tissue damage [46]. Together, these processes form a tangled web of early disruptions that set the stage for chronic neuroinflammation and degeneration.

#### 2.1.3. Cell Membrane Damage

Cell membrane damage is a key element of primary damage in CNS injuries, occurring as a direct result of the mechanical forces applied during the initial injury [41]. The cell membrane, or plasma membrane, is a lipid bilayer that surrounds the cell and controls the movement of substances into and out of the cell. It is essential for maintaining cellular homeostasis, signaling, and communication. When the CNS is injured, mechanical forces can cause physical disruption of the cell membrane, leading to a series of harmful events [11].

One of the immediate effects of cell membrane damage is the loss of ion balance. The cell membrane contains various ion channels and pumps that regulate the concentrations of ions such as sodium (Na+), potassium (K+), and calcium (Ca2+) across the membrane [47,48]. Damage to the membrane can cause dysfunction of these channels and pumps, resulting in an uncontrolled influx of ions [48,49]. This influx can lead to rapid depolarization of the cell membrane, initiating excitotoxicity, a process where excessive neuronal activity causes cellular damage and death. Calcium overload is a particularly harmful outcome of cell membrane disruption [50,51]. The uncontrolled entry of Ca2+ into the cell activates enzymes like proteases, lipases, and endonucleases, which degrade cellular components such as proteins, lipids, and DNA [52,53,54]. This degradation leads to cell death via necrosis or apoptosis. Additionally, excessive Ca2+ can impair mitochondrial function, generating ROS and causing further cellular damage [55,56].

The disruption of the cell membrane also causes the release of intracellular contents into the extracellular space. These contents are recognized by the immune system as DAMPs, which trigger inflammatory responses. This activation of immune cells, such as microglia and astrocytes, leads to the release of pro-inflammatory cytokines and chemokines, worsening the injury and contributing to secondary damage [57,58,59,60,61].

Lipid peroxidation is another significant consequence of cell membrane damage. The membrane is rich in polyunsaturated fatty acids, which are vulnerable to oxidative damage [62,63]. ROS can attack these fatty acids, leading to lipid peroxidation. This process produces toxic byproducts like malondialdehyde (MDA) and 4-hydroxynonenal (4-HNE), which can further damage cellular components and impair membrane function [64,65,66].

Furthermore, the physical disruption of the cell membrane can activate mechanosensitive ion channels, which worsen ion dysregulation and cellular damage. These channels open in response to mechanical stress, allowing ions to flow into the cell and contributing to the overall disruption of cellular homeostasis [67].

### 2.2. Secondary Damage Mechanisms

Secondary damage mechanisms in CNS injuries involve a series of biochemical and cellular processes that intensify the initial injury [68]. Keys among these are ionic imbalance and calcium overload, which significantly contribute to the spread of cellular damage and dysfunction. After the primary injury, disruption of the cell membrane and the malfunction of ion pumps and channels result in a loss of ion homeostasis, particularly impacting the levels of Na+, K+, and Ca2+ [69].

#### 2.2.1. Ionic Imbalance and Calcium Overload

The proper functioning of neurons depends on the precise regulation of ion concentrations across the cell membrane. Sodium and potassium ions are essential for maintaining the resting membrane potential and generating action potentials. Following an injury, the integrity of the cell membrane is disrupted, leading to an uncontrolled influx of Na+ and efflux of K+ [70]. This disturbance results in the sustained depolarization of neurons, which can cause the excessive release of neurotransmitters, especially glutamate. The overactivation of glutamate receptors, such as NMDA and AMPA receptors, further worsens ion dysregulation, creating a harmful cycle of neuronal excitation and damage [71,72,73].

Ca2+ serves as a critical secondary messenger in various cellular processes, including synaptic transmission, enzyme activation, and gene expression [74,75]. However, excessive intracellular Ca2+ can be detrimental. After CNS injury, the disruption of voltage-gated calcium channels and the overactivation of glutamate receptors result in a significant influx of Ca2+ into neurons [72]. This calcium overload triggers destructive pathways, first by activating enzymes such as proteases (e.g., calpains), lipases (e.g., phospholipase A2), and endonucleases, which degrade cellular proteins, lipids, and DNA, thereby disrupting cellular architecture and function and promoting cell death via necrosis or apoptosis [76,77,78,79]. Furthermore, high levels of Ca2+ are absorbed by mitochondria, impairing their function and reducing ATP production while increasing the production of reactive oxygen species (ROS), further damaging cellular components. Additionally, the sustained rise in intracellular Ca2+ amplifies glutamate release, worsening excitotoxicity and leading to widespread neuronal damage [80]. Calcium overload and ionic imbalance are central to secondary damage mechanisms in CNS injuries, disrupting cellular homeostasis, activating destructive enzymatic pathways, and contributing to mitochondrial dysfunction and excitotoxicity.

#### 2.2.2. Mitochondrial Dysfunction and Stress Reactions

Mitochondrial dysfunction is a key factor in secondary damage following CNS injuries, significantly contributing to cellular energy failure and oxidative stress [81,82,83]. Mitochondria are essential for ATP production, calcium buffering, and the regulation of apoptosis, and their structural and functional integrity is often compromised after an injury, initiating a cascade of detrimental effects [84,85,86]. Damage to the electron transport chain (ETC) and loss of mitochondrial membrane potential disrupt ATP synthesis, leading to energy failure that affects critical cellular processes, including ion pump function, neurotransmitter synthesis, and repair mechanisms [87]. Additionally, excessive calcium influx overwhelms the mitochondria’s buffering capacity, causing the mitochondrial permeability transition pore (mPTP) to open, which further collapses the membrane potential, impairs ATP production, and releases pro-apoptotic factors like cytochrome c [88,89,90].

Furthermore, dysfunctional mitochondria generate excessive ROS, which damage cellular components like lipids, proteins, and DNA, exacerbating oxidative stress and triggering inflammatory pathways [14,87,91]. In response to injury, mitochondria activate stress reactions such as the unfolded protein response (UPR), which attempts to restore protein homeostasis by enhancing chaperone expression and degrading misfolded proteins. However, prolonged activation of the UPR can ultimately lead to apoptosis [92,93,94]. Mitophagy, a process of selective autophagy that removes damaged mitochondria, is activated to prevent the spread of dysfunction, but excessive activation can deplete the mitochondrial pool, further impairing cellular energy supply [95,96,97]. Ultimately, mitochondrial dysfunction and stress responses play a central role in secondary damage following CNS injuries, leading to energy failure, oxidative stress, and apoptosis, which further exacerbate cellular injury [98].

#### 2.2.3. Excitability, Toxicity, and Chronic Inflammation

Excitotoxicity and chronic inflammation are interconnected secondary damage mechanisms that play a major role in the progression of CNS injuries, involving the overactivation of neuronal signaling pathways and prolonged immune responses, both of which lead to extensive tissue damage [99,100]. Excitotoxicity refers to the pathological process in which neurons are damaged or killed by the excessive activation of glutamate receptors, particularly NMDA and AMPA receptors [72]. After a CNS injury, the excessive release of glutamate and the failure of its reuptake mechanisms lead to sustained activation of these receptors, causing ion dysregulation [52,76]. This process disrupts synaptic transmission and plasticity, impairing neural communication and contributing to functional deficits [101,102,103].

Chronic inflammation, on the other hand, is a prolonged immune response initiated by the initial injury, resulting in the activation of microglia and astrocytes, which release pro-inflammatory cytokines (such as IL-1β, TNF-α) and chemokines [66,104,105,106,107]. While this response is initially protective, its continued activation leads to neuronal damage, as pro-inflammatory cytokines and ROS produced by activated glial cells directly harm neurons and oligodendrocytes, resulting in demyelination and axonal injury [108,109,110]. Chronic inflammation also contributes to the formation of a glial scar, which acts as both a physical and chemical barrier to axonal regeneration and repair [111,112]. Moreover, sustained inflammation worsens BBB disruption, allowing further infiltration of immune cells and harmful substances into the CNS, thereby compounding the damage. Together, excitotoxicity and chronic inflammation significantly contribute to secondary damage in CNS injuries, perpetuating a cycle of neuronal death and tissue dysfunction [113,114].

## 3. Neurodegenerative Pathways

Neurodegenerative diseases are characterized by the gradual decline in neuronal structure and function, often accompanied by the buildup of abnormal protein aggregates. Key proteins implicated in these diseases include Tau, amyloid-beta (Aβ), TAR DNA-binding protein 43 (TDP-43), and α-synuclein, each contributing in unique ways to the development of different neurodegenerative disorders.

### 3.1. Role of Tau Protein, Aβ Plaques, TDP-43, and α-Synuclein Deposits

Tau is a microtubule-associated protein that plays a crucial role in stabilizing microtubules within neurons, which is vital for maintaining cellular structure and facilitating intracellular transport [115,116]. In neurodegenerative conditions such as Alzheimer’s disease (AD) and frontotemporal dementia (FTD), Tau undergoes hyperphosphorylation, causing it to detach from microtubules and aggregate into neurofibrillary tangles (NFTs). This disruption destabilizes microtubules, impairs axonal transport, and results in neuronal dysfunction and death [117,118]. The spread of Tau pathology across different brain regions is thought to be linked to disease progression. Amyloid-beta (Aβ), a peptide derived from amyloid precursor protein (APP), accumulates in the form of extracellular plaques in AD due to abnormal APP cleavage [119,120]. Aβ plaques are neurotoxic, disrupting synaptic function and contributing to cognitive decline such as synaptic dysfunction, oxidative stress, and inflammation [121,122].

Aβ plaque accumulation is a defining feature of AD and is believed to trigger a chain of pathological events, including Tau hyperphosphorylation and NFT formation. TAR DNA-binding protein 43 (TDP-43), a protein involved in RNA processing, becomes mislocalized in diseases such as amyotrophic lateral sclerosis (ALS) and FTD, forming cytoplasmic aggregates that disrupt RNA metabolism and lead to neuronal dysfunction and death [117,123]. Mutations in the TARDBP gene, which encodes TDP-43, are associated with familial ALS and FTD, highlighting its crucial role in neuronal health [124,125]. α-Synuclein, a presynaptic protein implicated in Parkinson’s disease (PD) and other synucleinopathies, aggregates into Lewy bodies and Lewy neurites, disrupting neuronal function [126,127]. Although the exact role of α-synuclein in neurodegeneration remains incompletely understood, it is believed to contribute to impaired synaptic vesicle trafficking, mitochondrial dysfunction, and oxidative stress [126,128]. Mutations and gene duplications in SNCA, the gene encoding α-synuclein, are linked to familial PD [127]. The abnormal aggregation of Tau, Aβ plaques, TDP-43, and α-synuclein plays a critical role in the pathogenesis of various neurodegenerative diseases. These aggregated proteins disrupt essential cellular processes, leading to neuronal dysfunction and death.

Genetic background significantly influences disease progression and treatment efficacy. For instance, APOE4 allele carriers exhibit accelerated amyloid-beta accumulation and reduced response to anti-Aβ therapies in Alzheimer’s disease [119,120], while TARDBP mutations drive TDP-43 proteinopathy in familial ALS, necessitating gene-targeted approaches [124,125]. Similarly, SNCA mutations in Parkinson’s disease may require α-synuclein-targeted interventions [126,127]. Genetic screening thus holds potential for stratifying patients and optimizing therapeutic outcomes.

### 3.2. Mechanisms Leading to Neurodegeneration

Neurodegeneration is a multifaceted process involving various interconnected mechanisms that progressively lead to the loss of neuronal structure and function [129,130]. Key mechanisms include protein misfolding and aggregation, synaptic dysfunction, mitochondrial dysfunction, oxidative stress, and neuroinflammation. These processes collectively contribute to the development of neurodegenerative diseases [131]. A defining feature of these disorders is the accumulation of misfolded proteins that form toxic aggregates, disrupting cellular homeostasis [131,132]. For example, in Alzheimer’s disease (AD), the misfolding and aggregation of amyloid-beta (Aβ) and Tau lead to the formation of plaques and tangles that interfere with synaptic function, impair axonal transport, and ultimately cause neuronal death [133,134,135]. In Parkinson’s disease (PD), α-synuclein aggregates into Lewy bodies, which similarly disrupt synaptic vesicle trafficking and mitochondrial function [136,137].

Synaptic dysfunction, an early hallmark of neurodegeneration, often worsens due to the loss of synaptic proteins, impaired neurotransmitter release, and disrupted signaling pathways, contributing to cognitive and motor deficits. This synaptic dysfunction typically precedes neuronal death and plays a significant role in the clinical manifestations of these diseases [138,139]. Mitochondrial dysfunction further exacerbates neurodegeneration, as mitochondria are critical for energy production, calcium buffering, and apoptosis regulation [140,141]. In both PD and AD, mitochondrial function is compromised by protein aggregates, leading to energy failure, increased oxidative stress, and the activation of cell death pathways. Oxidative stress, resulting from an imbalance between reactive oxygen species (ROS) production and antioxidant defenses, is a major contributor to neuronal damage [66,142,143,144,145]. In amyotrophic lateral sclerosis (ALS), oxidative stress is linked to the toxicity of TDP-43 aggregates, which impair antioxidant defenses and increase ROS levels [146,147].

Neuroinflammation is another key driver of neurodegeneration, involving the activation of microglia and astrocytes, which release pro-inflammatory cytokines and chemokines [61,66,148,149]. While inflammation may initially serve a protective role, its chronic activation exacerbates neuronal damage. In both AD and PD, protein aggregates such as Aβ plaques, Tau tangles, and α-synuclein activate microglia, which release inflammatory mediators that further contribute to neurodegeneration [61,150]. In conclusion, neurodegeneration is driven by a combination of protein misfolding and aggregation, synaptic dysfunction, mitochondrial dysfunction, oxidative stress, and neuroinflammation, all of which disrupt cellular homeostasis and contribute to neuronal dysfunction and death.

## 4. Inflammatory Response

The inflammatory response in the CNS is a complex, tightly controlled process that plays a significant role in both the progression and repair of CNS injuries. Following injury, the activation of glial cells, such as microglia and astrocytes, triggers the release of cytokines and chemokines [151,152]. These signaling molecules facilitate communication between immune cells, neurons, and glial cells. Cytokines like interleukins and TNF, along with chemokines such as CCL2 and CXCL1, govern the recruitment and activation of immune cells, thereby contributing to the inflammatory response in the CNS [110,153]. While the inflammatory response initially serves a protective role, it can become harmful if it is prolonged or excessive [152].

In the acute phase, inflammation aids in clearing debris, promoting tissue repair, and containing injury [154]. However, chronic inflammation can worsen damage, resulting in neuronal death, gliosis, and further disruption of CNS function [155,156]. The persistent activation of glial cells and the continuous release of pro-inflammatory mediators can interfere with normal synaptic function, leading to cognitive and motor impairments [157]. Therefore, while inflammation is essential for CNS injury repair, its regulation is crucial, as dysregulated inflammation can exacerbate secondary damage and negatively impact the overall prognosis of CNS injuries [113].

### 4.1. Release of Cytokines and Chemokines

Cytokines are small proteins essential for regulating immune responses, inflammation, and hematopoiesis, particularly during CNS injury [158,159]. Pro-inflammatory cytokines, such as IL-1β, are rapidly released following CNS injury, mainly by activated microglia and astrocytes [160]. IL-1β exacerbates inflammation by promoting the expression of additional pro-inflammatory cytokines, adhesion molecules, and chemokines, which can contribute to blood–brain barrier disruption and neuronal apoptosis [161,162]. TNF-α, produced by microglia, astrocytes, and infiltrating immune cells, has both protective and harmful effects. It can induce apoptosis in neurons and oligodendrocytes, worsen blood–brain barrier damage, and amplify the inflammatory response [163,164].

IL-6, produced by microglia, astrocytes, and neurons, exerts multiple effects, including promoting inflammation, regulating immune cell differentiation, and influencing neuronal survival. Elevated IL-6 levels are linked to chronic inflammation and neurodegenerative diseases [165]. On the other hand, anti-inflammatory cytokines such as IL-10 and Transforming Growth Factor-β (TGF-β) help resolve inflammation and limit tissue damage. IL-10 suppresses the production of pro-inflammatory cytokines and microglial activation, while TGF-β promotes tissue repair and reduces microglial activation [107,166,167].

Chemokines, another group of signaling molecules, direct immune cell migration to sites of injury. For example, CCL2 (MCP-1) attracts monocytes, macrophages, and T cells to the CNS, supporting neuroprotection and inflammation [168]. Other chemokines, including CCL5 (RANTES), CXCL8 (IL-8), and CXCL10 (IP-10), recruit various immune cells, such as T cells, neutrophils, and monocytes, to regulate the inflammatory response [169,170]. Overall, cytokines and chemokines play a central role in orchestrating immune cell activation and recruitment during CNS injury, thereby influencing both the inflammatory response and tissue repair processes.

### 4.2. Activation of Astrocytes and Microglia

The activation of astrocytes and microglia is a key feature of the inflammatory response following CNS injury, with these glial cells playing crucial roles in both initiating and resolving inflammation, as well as contributing to tissue repair and neurodegeneration [171,172]. Astrocytes, which are the most abundant glial cells in the CNS, undergo reactive astrogliosis after injury, a process characterized by both morphological and functional changes [173,174]. These cells become hypertrophic, displaying increased size and number of processes, and upregulate proteins such as glial fibrillary acidic protein (GFAP) [175]. Functionally, reactive astrocytes exhibit both pro-inflammatory and anti-inflammatory roles. They secrete pro-inflammatory cytokines (e.g., IL-1β, TNF-α, IL-6) and chemokines (e.g., CCL2, CXCL10), amplifying the inflammatory response, while also producing ROS and nitric oxide (NO), which contribute to tissue damage [176]. However, they also release anti-inflammatory cytokines (e.g., IL-10, TGF-β) and neurotrophic factors (e.g., BDNFs) that aid in tissue repair and promote neuroprotection [177]. In the case of severe injuries, reactive astrocytes form a glial scar that, while containing inflammation and protecting surrounding tissue, can impede axonal regeneration and hinder functional recovery [178]. Microglia, the resident immune cells of the CNS, also become activated after injury, transitioning from their resting ramified state to an activated amoeboid form with retracted processes and enlarged cell bodies [171].

### 4.3. Recruitment of Circulating Immune Cells

The recruitment of circulating immune cells to the CNS is a critical component of the inflammatory response following injury, and this process is tightly regulated by adhesion molecules, chemokines, and cytokines [153,179]. Adhesion molecules are essential for the movement of immune cells from the bloodstream to the injured area, facilitating their attachment, rolling, adhesion, and transmigration across the BBB [18,180]. Selectins such as P-selectin and E-selectin, found on endothelial cells, mediate the initial attachment and rolling of leukocytes along the vessel walls, while integrins like LFA-1 and VLA-4 on leukocytes bind to ICAM-1 and VCAM-1 on endothelial cells to ensure firm adhesion, enabling the immune cells to migrate through the endothelial barrier [181,182]. Chemokines are signaling proteins that direct immune cells to injury sites, and they are produced by resident CNS cells, including astrocytes, microglia, and endothelial cells, in response to injury. For example, CCL2 (MCP-1) attracts monocytes, macrophages, and T cells, while CCL5 (RANTES) recruits T cells, monocytes, and eosinophils [183,184]. CXC chemokines like CXCL8 (IL-8) and CXCL10 (IP-10) further assist in the recruitment of neutrophils and T cells [185]. Cytokines also play a key role in immune cell recruitment by modulating the expression of adhesion molecules and chemokines. Pro-inflammatory cytokines such as IL-1β and TNF-α enhance the expression of these molecules, promoting immune cell infiltration, while anti-inflammatory cytokines like IL-10 and TGF-β help resolve inflammation by inhibiting their expression [186]. Collectively, these molecular signals coordinate the migration of immune cells to the site of CNS injury, facilitating the inflammatory response.

## 5. CNS Repair Mechanisms

The CNS, which includes the brain and spinal cord, is an exceptionally complex and sensitive structure responsible for regulating most bodily functions and mental processes. Unlike other tissues in the body, the CNS has limited capacity for self-repair following injury or disease [187]. However, recent advancements have revealed that the CNS does possess some repair mechanisms, although these are constrained. Two key processes involved in CNS repair are synaptogenesis, the creation of new synapses, and neural repair mechanisms [2]. These processes are critical for restoring function after injury, addressing neurodegenerative diseases, and preserving cognitive health.

### 5.1. Formation of New Synapses

Synaptogenesis, the process of forming new synapses, is essential for supporting learning, memory, and recovery after injury in the CNS [188,189]. While synaptogenesis occurs throughout life, it is most pronounced during early development as the brain rapidly establishes neural circuits [189,190]. In adulthood, the process is more limited but can be reactivated in response to injury, learning, or environmental stimuli. It is a key aspect of neuroplasticity, which enables the brain to reorganize by forming new neural connections. This plasticity involves the growth of dendritic spines and the creation of new synaptic contacts, with synaptic strength shaped by experience [189]. Several molecular signals, including BDNFs, are crucial for promoting synapse growth and differentiation, while neuroligins and neurexins ensure proper synaptic function by mediating the adhesion and alignment of pre- and post-synaptic neurons [191,192].

Glial cells, especially astrocytes and microglia, also play roles in synaptogenesis, with astrocytes releasing factors that encourage synapse formation and microglia pruning unnecessary or dysfunctional synapses to refine neural circuits [193]. External factors, such as physical exercise, cognitive training, and enriched environments, further enhance synaptogenesis by increasing neuronal activity and stimulating the release of growth factors, thereby promoting a conducive environment for new synapse formation [194,195]. In the aftermath of CNS injuries like stroke or traumatic brain injury, synaptogenesis plays a crucial role in recovery [196]. Damaged neurons attempt to rebuild connections by forming new synapses, a process referred to as reactive synaptogenesis [189]. However, this process is often incomplete or inefficient, leading to ongoing functional impairments. Enhancing synaptogenesis through therapeutic strategies such as pharmacological interventions or neuromodulation presents a promising avenue for improving recovery outcomes.

### 5.2. Neural Repair Processes

Neural repair refers to the processes by which the CNS attempts to restore function following injury or disease [197]. Unlike the peripheral nervous system (PNS), which has a greater regenerative capacity, the CNS faces several significant challenges, including inhibitory molecules in the extracellular matrix, limited neuronal regeneration, and the formation of glial scars [198,199]. Despite these obstacles, the CNS retains some intrinsic repair mechanisms that can support recovery [200]. One such mechanism is axonal regeneration, where axons, long projections of neurons responsible for transmitting signals, attempt to regrow after injury [2,201]. However, this process is often hindered by inhibitory factors like myelin-associated proteins (such as Nogo-A) and chondroitin sulfate proteoglycans (CSPGs) [202,203]. Research is actively exploring ways to overcome these barriers, including blocking inhibitory signals or stimulating growth-promoting genes.

Another vital process in neural repair is neurogenesis, the generation of new neurons in specific regions of the adult brain, such as the hippocampus and subventricular zone. Though limited, neurogenesis plays an essential role in learning, memory, and recovery from injury, and current research is focused on enhancing it through methods such as promoting BDNF expression or utilizing stem cell therapy [191,204]. Glial cells, including astrocytes and microglia, also participate in CNS repair. While they contribute to the formation of glial scars that can impede recovery, they also release growth factors and cytokines that aid tissue repair and modulate inflammation [205,206,207]. A significant challenge in CNS repair is balancing the beneficial and detrimental effects of glial cells. Inflammation plays a dual role: acute inflammation is necessary to clear damaged tissue and facilitate repair, while chronic inflammation can exacerbate damage [208,209].

Lastly, remyelination, the restoration of the protective myelin sheath around axons, is essential for proper signal transmission and preventing further degeneration [208,210]. Oligodendrocyte precursor cells (OPCs) are critical for remyelination, and efforts are underway to enhance their differentiation to support recovery in CNS injuries and diseases such as multiple sclerosis [211,212,213]. Despite these repair mechanisms, the CNS faces substantial challenges in achieving functional recovery, including intrinsic limitations, as mature neurons have significantly lower regenerative capacities compared to their developmental counterparts [214]. Moreover, extrinsic barriers such as inhibitory molecules and physical obstacles like glial scars hinder the repair process [215]. The complexity of neural circuits also complicates recovery, as effective repair requires not just the regeneration of individual neurons but also the restoration of intricate neural networks necessary for proper function [216].

To address these limitations, various therapeutic strategies are being explored to enhance the regenerative potential of the CNS. Pharmacological interventions, including BDNF mimetics and Nogo receptor antagonists, have shown promise in preclinical studies for promoting neurogenesis, synaptogenesis, and axonal regeneration [217,218,219]. Stem cell therapies, utilizing neural stem cells and induced pluripotent stem cells (iPSCs), offer the potential to replace lost neurons and support the repair process [220]. Gene therapy, involving the delivery of genes that promote growth factors or inhibit inhibitory signals, is also under investigation as a potential approach [221,222]. Additionally, rehabilitation techniques such as physical therapy, cognitive training, and neurostimulation methods like transcranial magnetic stimulation (TMS) are being explored for their potential to stimulate neuroplasticity and assist in CNS repair [223].

## 6. Role of Microglia in Injury and Repair

Microglia are the resident immune cells of the CNS, playing a critical role in maintaining homeostasis, responding to injury, and supporting repair processes [224,225]. These cells are highly versatile and can rapidly adjust their phenotype and functions in response to environmental cues. In the context of CNS injury and repair, microglia exhibit a dual role, where they can either exacerbate damage or aid in recovery, depending on their activation state. Microglia are typically classified into two main phenotypes: M1-like (pro-inflammatory) and M2-like (anti-inflammatory). Their phagocytic function is also vital for clearing cellular debris and supporting tissue repair.

### 6.1. M1-like Microglia and Pro-Inflammatory Responses

M1-like microglia are classically activated immune cells that play a crucial role in initiating pro-inflammatory responses within the CNS. These cells are primarily involved during the early stages of injury or infection, where they help defend the CNS by combating pathogens and damaged cells [171,226]. However, the sustained or excessive activation of M1-like microglia can lead to additional tissue damage and contribute to the progression of neurodegenerative diseases [150,227]. M1-like microglia are activated by various signals such as lipopolysaccharide (LPS), IFN-γ, and DAMPs released from injured cells [228,229]. Upon activation, they undergo morphological changes, transitioning from a branched, ramified structure to an amoeboid form, enabling them to migrate to injury sites. At the injury location, these microglia secrete pro-inflammatory cytokines such as TNF-α, IL-1β, and IL-6, further amplifying the inflammatory response and attracting additional immune cells [230,231]. They also produce ROS and NO, which are toxic to pathogens and damaged cells, but excessive production of these molecules can damage healthy neurons and glial cells, leading to secondary injury [61,171].

Additionally, M1-like microglia express major histocompatibility complex (MHC) class II molecules, allowing them to present antigens to T cells and activate adaptive immune responses. While this is essential for fighting infections, it can also contribute to autoimmune responses within the CNS [232]. In acute injuries such as stroke or traumatic brain injury, M1-like microglia are among the first responders, clearing debris and attempting to limit damage. However, their pro-inflammatory activity can exacerbate tissue injury, promote neuroinflammation, and lead to neuronal death [113,233]. In chronic conditions like Alzheimer’s disease or multiple sclerosis, sustained M1-like activation contributes to neurodegeneration. As a result, targeting M1-like microglia has become a potential therapeutic approach for CNS injuries and diseases, with strategies including anti-inflammatory drugs that inhibit pro-inflammatory cytokines or ROS production, as well as therapies aimed at modulating microglial activation to shift them from an M1-like to an M2-like phenotype, promoting repair and reducing inflammation [226,230,234,235,236].

### 6.2. M2-like Microglia and Anti-Inflammatory Responses

M2-like microglia are alternatively activated cells that play a crucial role in promoting anti-inflammatory responses and supporting tissue repair in the CNS. These microglia are primarily involved in the later stages of injury resolution, contributing to the restoration of CNS homeostasis [237,238]. Known for their ability to suppress inflammation, clear debris, and enhance neuronal survival and regeneration, M2-like microglia are essential for effective recovery. They are activated by signals such as IL-4, IL-10, and transforming growth factor-beta (TGF-β), which promote a transition from a pro-inflammatory to an anti-inflammatory state [184]. This phenotypic shift enables them to foster tissue repair. M2-like microglia secrete anti-inflammatory cytokines such as IL-10 and TGF-β, helping to dampen inflammation and support healing. They also release growth factors like insulin-like growth factor-1 (IGF-1) and BDNFs, which promote neuronal survival and synaptic plasticity [239,240]. These cells are also involved in extracellular matrix remodeling by producing enzymes like MMPs and tissue inhibitors of metalloproteinases (TIMPs), which assist in tissue repair [241]. M2-like microglia additionally modulate immune responses and promote inflammation resolution through the expression of molecules such as arginase-1 and chitinase-like proteins. In CNS injuries like spinal cord injury or stroke, the transition from M1-like to M2-like microglia is crucial for functional recovery, as they aid in clearing debris, reducing neuroinflammation, and creating a favorable environment for neuronal regeneration [242,243]. Enhancing the activity of M2-like microglia offers a promising therapeutic strategy for CNS injuries and diseases, with approaches like cytokine therapy using IL-4 or IL-10 to promote the M2-like phenotype, as well as stem cell therapies to facilitate tissue repair by inducing M2-like activation [244] (Figure 2).

### 6.3. Phagocytic Activity and Tissue Repair

Phagocytosis is a vital function of microglia, involving the engulfment and digestion of cellular debris, pathogens, and damaged neurons. This process is crucial for maintaining homeostasis in the CNS and supporting tissue repair following injury [245,246]. Microglia identify targets for phagocytosis using pattern recognition receptors (PRRs), such as toll-like receptors (TLRs) and scavenger receptors, which bind to DAMPs, pathogen-associated molecular patterns (PAMPs), or apoptotic cells, initiating the engulfment process. Once internalized, the targets are degraded within phagolysosomes, and the waste products are expelled, clearing debris and preventing the accumulation of toxic substances [150,171,247]. After CNS injury, microglia rapidly migrate to the damaged area to phagocytose dead cells and debris, playing a crucial role in preventing secondary damage and creating a favorable environment for repair [248].

Furthermore, microglia engage in synaptic pruning, eliminating unnecessary or dysfunctional synapses to refine neural circuits during both developmental stages and after injury [249,250,251]. They also contribute to tissue remodeling by clearing debris and restructuring the extracellular matrix, collaborating with other glial cells like astrocytes and oligodendrocytes to support repair processes. Enhancing the phagocytic activity of microglia presents a potential therapeutic approach for CNS injuries and diseases [246]. This could involve promoting phagocytosis through compounds that increase the expression of phagocytic receptors or stimulate microglial activity to improve debris clearance and tissue regeneration [252]. In conditions like Alzheimer’s disease, where microglial phagocytosis is impaired, restoring this function may help clear pathological protein aggregates such as amyloid-beta, offering a promising therapeutic strategy [253]. Understanding the complex roles of microglia in injury and repair provides valuable insights into potential therapeutic strategies for CNS injuries and neurodegenerative diseases [150] (Figure 3).

## 7. Therapeutic Implications

The CNS has a limited capacity for regeneration, posing a major challenge in developing effective therapeutic strategies for CNS injuries and diseases. Despite this, recent advancements in neuroscience have revealed several promising approaches aimed at promoting recovery and protecting neural tissue. These strategies include targeting inflammatory pathways, enhancing the CNS’s intrinsic repair mechanisms, and developing neuroprotective treatments.

### 7.1. Targeting Inflammatory Pathways

Inflammation in the CNS plays a complex role, serving both as a vital process for repair and a potential contributor to further damage [209,254]. While acute inflammation is essential for clearing debris and initiating the recovery process, chronic or excessive inflammation can worsen tissue damage and accelerate neurodegeneration [148,255]. Consequently, targeting inflammatory pathways has become a key therapeutic approach to mitigate the detrimental effects of neuroinflammation while preserving its beneficial functions.

Several critical inflammatory pathways are involved in CNS inflammation, including cytokine and chemokine signaling, TLR signaling, and the NLRP3 inflammasome. Pro-inflammatory cytokines, such as TNF-α, IL-1β, and IL-6, released by activated microglia and astrocytes, can cause neuronal damage and disrupt the blood–brain barrier (BBB) [256,257,258]. Chemokines like CXCL12 and CCL2 recruit immune cells to the site of injury, amplifying the inflammatory response but potentially contributing to tissue damage, particularly in autoimmune conditions like multiple sclerosis [259]. TLRs, which detect DAMPs and PAMPs, initiate inflammatory responses that are beneficial in acute injury but harmful in chronic conditions. The NLRP3 inflammasome, which produces IL-1β and IL-18, is dysregulated in neurodegenerative diseases like Alzheimer’s and Parkinson’s [260,261].

Therapeutic strategies targeting these inflammatory pathways include anti-cytokine treatments, such as monoclonal antibodies and small-molecule inhibitors, designed to reduce neuroinflammation in diseases like Alzheimer’s and stroke. Chemokine receptor antagonists are being explored to block immune cell recruitment, potentially offering a treatment for multiple sclerosis and other autoimmune disorders [262,263]. Modulating TLR activity with TLR4 antagonists has shown neuroprotective effects in models of traumatic brain injury and stroke [264,265,266]. Furthermore, NLRP3 inflammasome inhibitors like MCC950 have demonstrated promise in reducing neuroinflammation and improving outcomes in animal models of neurodegenerative diseases [264,266,267].

Despite the potential of these strategies, several challenges remain, including the need for selective inhibition of harmful inflammation without impairing beneficial immune responses, the risk of systemic side effects, and the complexity of inflammatory signaling networks.

### 7.2. Enhancing Repair Mechanisms

The CNS has a limited capacity for self-repair, but recent advancements have revealed several mechanisms that could be leveraged to promote recovery after injury or disease. Enhancing these repair processes is a critical therapeutic approach to improving outcomes in patients with CNS disorders [130,187].

Key repair mechanisms in the CNS include neurogenesis, synaptogenesis, axonal regeneration, and remyelination [268,269,270]. Neurogenesis, the process by which new neurons are generated from neural stem cells in areas such as the hippocampus and subventricular zone, offers potential for replacing lost neurons and restoring function [191,204]. Synaptogenesis, or the formation of new synapses, is crucial for repairing neural circuits and connectivity following injury and can be enhanced through environmental enrichment or pharmacological interventions [189,271,272]. Axonal regeneration is particularly challenging in the CNS due to inhibitory factors in the extracellular matrix and the limited regenerative potential of mature neurons, making it a major research focus [201]. Remyelination, which involves the restoration of myelin sheaths around axons, is essential for efficient signal transmission and protecting neurons from further degeneration, with OPCs playing a central role in this process [273,274].

Several therapeutic strategies are being explored to enhance these repair mechanisms. Stem cell therapy, utilizing neural stem cells or induced pluripotent stem cells (iPSCs), shows promise for replacing lost neurons and supporting recovery. Growth factors like BDNF, IGF-1, and glial cell line-derived neurotrophic factors (GDNFs) promote neuron survival and growth, delivered via gene therapy or biomaterials [275,276,277,278,279]. Modulating the extracellular matrix by inhibiting inhibitory molecules such as chondroitin sulfate proteoglycans (CSPGs) and myelin-associated proteins like Nogo-A can facilitate axonal regeneration [280,281,282]. Moreover, pharmacological agents targeting neurogenesis, synaptogenesis, or remyelination are being developed, such as drugs that stimulate the Wnt/β-catenin pathway for neurogenesis or clementine for remyelination [283]. Despite the promise of these strategies, enhancing CNS repair remains a complex challenge, as it requires overcoming significant obstacles, such as the inhibitory environment, the limited regenerative capacity of neurons, and the need to restore intricate neural circuits.

### 7.3. Potential for Neuroprotective Strategies

Neuroprotective strategies aim to prevent or reduce neuronal damage, thereby preserving function, and are especially important in acute CNS injuries, such as stroke and traumatic brain injury, as well as in chronic neurodegenerative diseases [284,285]. These strategies focus on several key mechanisms, including reducing excitotoxicity, combating oxidative stress, preventing apoptosis, and safeguarding mitochondrial function [286,287]. Excitotoxicity, which is caused by the excessive release of glutamate and overstimulation of glutamate receptors, is a primary driver of neuronal cell death, making its inhibition a central neuroprotective approach [288]. Oxidative stress, resulting from excessive reactive oxygen species (ROS), plays a significant role in neuronal damage, and antioxidants can help neutralize these ROS, offering protection to neurons [289]. Apoptosis, or programmed cell death, is another major pathway for neuronal loss in CNS injuries and diseases, and blocking apoptotic pathways may help preserve neurons [290]. Mitochondrial dysfunction is commonly observed in CNS injuries and neurodegenerative conditions, so protecting mitochondrial function is crucial for enhancing neuronal survival [291,292].

Therapeutic approaches designed to achieve these goals include the use of glutamate receptor antagonists such as memantine, which blocks NMDA and AMPA glutamate receptors and has shown neuroprotective effects in conditions like stroke and traumatic brain injury [9,293]. Antioxidants like coenzyme Q10, vitamin E, and edaravone have been explored for their potential to reduce oxidative stress and protect neurons [294,295,296]. Caspase inhibitors targeting caspase-3 and caspase-9 have shown promise in preclinical studies by preventing apoptosis in CNS injuries [297,298]. Additionally, mitochondrial-targeted agents such as MitoQ, which reduce mitochondrial oxidative damage, are being investigated for their potential to offer neuroprotection [299].

### 7.4. Disease-Tailored Therapeutic Approaches

While common pathological mechanisms exist across central nervous system disorders, effective therapeutic interventions must be specifically tailored to address distinct disease processes. In Parkinson’s disease, characterized by the selective degeneration of dopaminergic neurons in the substantia nigra, stem cell-based approaches offer significant potential for targeted neuronal replacement [300]. Multiple sclerosis presents a fundamentally different challenge, with its autoimmune-mediated destruction of myelin, making immunomodulatory therapies such as anti-CD20 monoclonal antibodies or remyelinating agents like clemastine particularly valuable treatment options [301,302]. Alzheimer’s disease presents yet another therapeutic paradigm, where the coexistence of amyloid plaques and neurofibrillary tau tangles likely necessitates combination therapies that simultaneously target both pathological hallmarks [303,304]. These examples underscore the critical importance of developing precision medicine approaches that account for the unique neuropathological features of each disorder.

## 8. Conclusions

In conclusion, significant progress has been made in understanding CNS injury and repair, offering promising prospects for better patient outcomes in CNS-related disorders. The mechanisms of CNS injury, encompassing both primary and secondary damage, inflammation, and neurodegeneration, have been more clearly defined, paving the way for innovative therapeutic approaches. While acute inflammation plays a vital role in initiating repair, chronic inflammation can worsen damage, emphasizing the need for therapies that can specifically target inflammatory pathways. Repair processes such as neurogenesis, synaptogenesis, and remyelination, though limited, lay the groundwork for enhancing recovery through stem cell therapies, growth factors, and pharmacological treatments. Additionally, neuroprotective strategies aimed at mitigating excitotoxicity, oxidative stress, and mitochondrial dysfunction are essential for preserving neuronal function.

By targeting inflammatory pathways, enhancing repair processes, and developing neuroprotective therapies, researchers and clinicians can work together to restore function and improve the quality of life for individuals affected by CNS injuries and diseases. The progress made so far highlights the potential for transformative therapies, but ongoing efforts are essential to fully unlock the regenerative potential of the CNS and address the remaining challenges.

## Figures and Tables

**Figure 1 cells-14-00918-f001:**
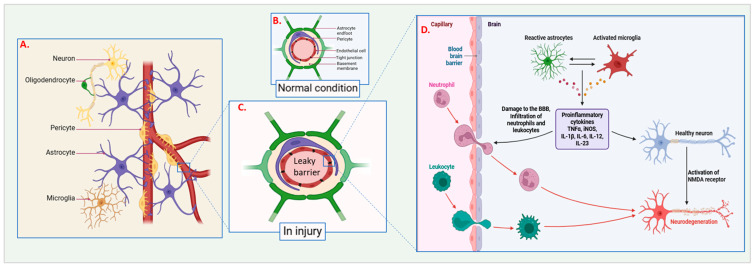
Cellular and molecular responses in the central nervous system (CNS) under normal and injured conditions: (**A**) Overview of major CNS cell types: Neurons, oligodendrocytes, pericytes, astrocytes, and microglia. (**B**) Normal conditions: Under typical physiological circumstances, astrocytes and pericytes help maintain the integrity of the blood–brain barrier (BBB), keeping its membrane structure intact. (**C**) Disrupted barrier in injury: Injury to the BBB results in its compromise, causing the barrier to become leaky and permitting harmful substances and cells to infiltrate the CNS. (**D**) Capillary and BBB components: Depicts the structure of capillaries, the blood–brain barrier, neuroglial cells, and leukocytes. In response to injury, astrocytes become reactive, and microglia are activated. This activation contributes to neural tissue damage, the infiltration of neutrophils and other leukocytes, and the release of pro-inflammatory cytokines such as TNF-α, IL-1β, IL-6, IL-12, and IL-23. The illustration contrasts a healthy neuron with intact NMDA receptors against the neurodegeneration resulting from inflammatory and cytotoxic processes following injury.

**Figure 2 cells-14-00918-f002:**
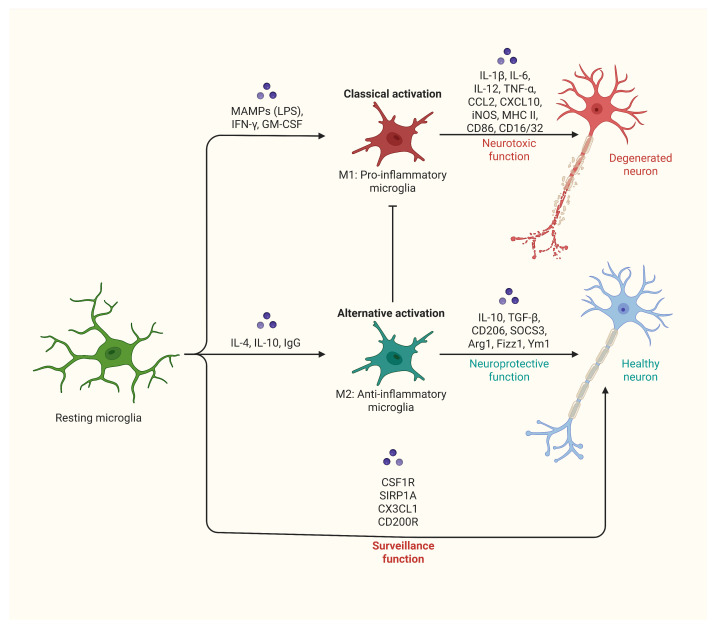
Activation states of microglia and their functional outcomes: **Classical activation (M1 phenotype):** Induced by MAMPs (e.g., LPS), IFN-γ, and GM-CSF. This state is characterized by the release of pro-inflammatory cytokines and mediators, including IL-1β, IL-6, IL-12, TNF-α, CCL2, CXCL10, iNOS, MHC II, CD86, and CD16/32. M1 activation is associated with neurotoxic effects, contributing to neuronal damage and degeneration. **Alternative activation (M2 phenotype):** This represents an anti-inflammatory response by microglia. In this state, microglia secrete anti-inflammatory cytokines such as IL-10 and TGF-β and express markers including Arg1 and Ym1, etc. M2 activation supports neuroprotective functions, helping to maintain healthy neurons. Additionally, it plays a role in surveillance activities mediated by CSF1R, SIRP1A, CX3CL1, and CD200R.

**Figure 3 cells-14-00918-f003:**
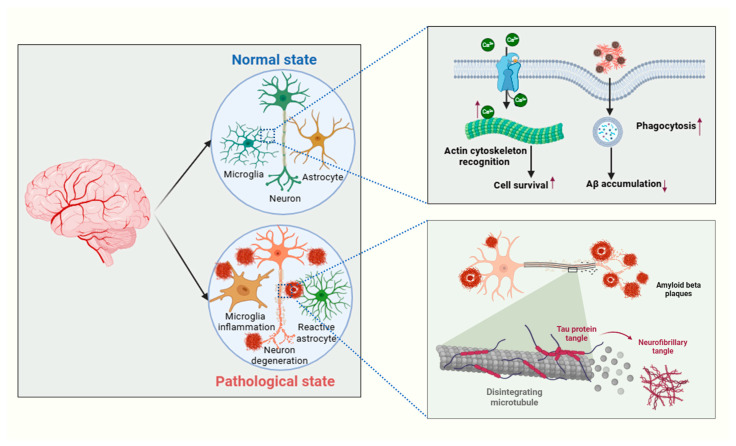
Microglial and astrocytic responses in normal and pathological states. Normal state: In the healthy CNS, microglia and astrocytes work together to maintain homeostasis. Neurons remain functional and undamaged. In response to inflammatory signals, microglia become activated, and astrocytes become reactive. This activation leads to neuronal degeneration and disrupts normal neural function. Microglia exhibit enhanced actin cytoskeleton recognition, promoting cell survival, increased phagocytosis, and improved clearance of amyloid-beta (Aβ). This results in a reduction in Aβ accumulation. However, pathological features such as tau protein tangles, disrupted microtubules, amyloid-beta plaques, neurofibrillary tangles, and necrotic tangles emerge, all of which contribute to the progression of neurodegenerative diseases.

## Data Availability

Not applicable.

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
