# Peer review of "Cellular and Molecular Interactions in CNS Injury: The Role of Immune Cells and Inflammatory Responses in Damage and Repair"

_cells, 2025, doi:10.3390/cells14120918_

Round 1

Reviewer 1 Report

Comments and Suggestions for Authors

The review article titled mechanistic insights into CNS damage and repair: cellular and molecular interactions following injury by Shukla etal. blends CNS injuries and repair mechanisms. The review has potential for high impact. Few major concerns for the authors to address:

  1. The authors mentioned primary and secondary damage to the CNS however these variables can interchange e.g ROS generated within an environment can cause damage to a distal membrane leading to increases in permeability. At the same time increase in permeability can result in increase in generation of free radicals. So, it is context dependent.
  2. It is not clear what causes the physical disruption of BBB, cells and blood vessels.
  3. How does microvascular damage occur as the authors mentioned that it causes primary damage following CNS injury affecting BBB.
  4. In Figure 2: M2 anti-inflammatory microglia block M1 and repair damage neuron to healthy neuron. The arrow directed to healthy neuron from M2 should be point to damaged neuron then another from damage to healthy neuron.

Author Response

Response to Reviewers

We thank all the reviewers for their insightful comments. Below, we provide a point-by-point response.

Reviewer 1

RC 1. The authors mentioned primary and secondary damage to the CNS however these variables can interchange e.g ROS generated within an environment can cause damage to a distal membrane leading to increases in permeability. At the same time increase in permeability can result in increase in generation of free radicals. So, it is context dependent.

Response: You raise an excellent point! We agree that the distinction between primary and secondary damage in the CNS is not always rigid, as certain pathological processes such as ROS generation and membrane permeability changes can act bidirectionally and reinforce one another. In response, we have revised Section 2 to explicitly acknowledge this context-dependent nature. Specifically, we added the following sentence on Lines 64–67.

RC 2. It is not clear what causes the physical disruption of BBB, cells and blood vessels.

Response: You're absolutely right to highlight this important gap. Our understanding of the precise mechanisms driving the initial physical disruption of the BBB, vasculature, and cellular structures in CNS injuries remains incomplete. In response to your comment, we have added a new subsection Section 2.1.b. Mechanical Forces and Blood-Brain Barrier Disruption During Injury (Lines 122–140). which provides mechanistic insight into the factors contributing to BBB breakdown. This new paragraph discusses the role of mechanical forces during trauma, as well as the downstream contribution of inflammatory mediators, MMPs, and oxidative stress in compromising BBB integrity.

RC 3. How does microvascular damage occur as the authors mentioned that it causes primary damage following CNS injury affecting BBB.

Response: Thank you for this insightful question. To clarify the mechanisms of microvascular damage contributing to primary injury and BBB disruption, we have expanded the relevant section to include a detailed explanation. These additions can be found in Lines 104–108 of the revised manuscript.

RC 4. In Figure 2: M2 anti-inflammatory microglia block M1 and repair damage neuron to healthy neuron. The arrow directed to healthy neuron from M2 should be point to damaged neuron then another from damage to healthy neuron.

Response: Thank you for pointing out this graphical inconsistency. We have revised Figure 2 to more accurately reflect the biological sequence of events.

Reviewer 2 Report

Comments and Suggestions for Authors Overall, the review is comprehensive but a bit too general, as each nervous injury or neurodegenerative disease is specific and more susceptible to one type of damage or another, which makes therapeutics less effective if using the same strategy for all.  For example, stem cell-based neuroregeneration could be better for Parkinson's disease (PD), as only dopaminergic neurons are affected in a focal area, whereas inflammation-inhibiting/immunomodulation could be more suitable for the treatment of multiple sclerosis (MS). Another consideration is the genetic background, as some of the gene mutations are involved in nervous injuries/diseases, such as the ApoE 4 gene in Alzheimer's disease (AD) and TDP-43 in Amyotrophic Lateral Sclerosis (ALS).  One small issue is that the colour of some words in Figure 1D should be darker and have more contrast with the background, which will make it clearer to read.

Author Response

Response to Reviewers

We thank all the reviewers for their insightful comments. Below, we provide a point-by-point response.

Reviewer 2

RC 1. Overall, the review is comprehensive but a bit too general, as each nervous injury or neurodegenerative disease is specific and more susceptible to one type of damage or another, which makes therapeutics less effective if using the same strategy for all. For example, stem cell-based neuroregeneration could be better for Parkinson's disease (PD), as only dopaminergic neurons are affected in a focal area, whereas inflammation-inhibiting/immunomodulation could be more suitable for the treatment of multiple sclerosis (MS).

Response: We appreciate this constructive feedback. In response, we have revised the manuscript to include a new subsection titled 7.4 Disease-Tailored Therapeutic Approaches (Lines 700–713). This section elaborates on the importance of developing therapeutic strategies that are tailored to the specific pathophysiological features of different CNS disorders.

RC 2. Another consideration is the genetic background, as some of the gene mutations are involved in nervous injuries/diseases, such as the ApoE 4 gene in Alzheimer's disease (AD) and TDP-43 in Amyotrophic Lateral Sclerosis (ALS).

Response: We appreciate this constructive feedback. In response, we have added content discussing the role of genetic factors in modulating the CNS injury response and individual susceptibility. Specific examples now included in the text highlight the involvement of ApoE4 and TDP-43. These additions can be found in Lines 291–297 of the revised manuscript.

RC 3. One small issue is that the colour of some words in Figure 1D should be darker and have more contrast with the background, which will make it clearer to read.

Response: Thank you for the helpful suggestion. We have updated Figure 1D to enhance text contrast and improve overall readability, ensuring that all labels and annotations are now clearly visible.

Reviewer 3 Report

Comments and Suggestions for Authors

This is a review article describing tissue damage after CNS injury, with focus on the role of inflammatory reaction to injury, and immune cells. It is well illustrated. The authors focus also on the neurodegenerative processes following injury. I would suggest that the title, which is now rather generic, should reflect the focus of the article the role of immune cells and inflammation after injury.

Author Response

RC 1. This is a review article describing tissue damage after CNS injury, with focus on the role of inflammatory reaction to injury, and immune cells. It is well illustrated. The authors focus also on the neurodegenerative processes following injury. I would suggest that the title, which is now rather generic, should reflect the focus of the article the role of immune cells and inflammation after injury.

Response: We appreciate the suggestion and have revised the title to better reflect the focus on immune and inflammatory responses after CNS injury. The new title is:
Cellular and Molecular Interactions in CNS Injury: The Role of Immune Cells and Inflammatory Responses in Damage and Repair

Round 2

Reviewer 1 Report

Comments and Suggestions for Authors

I do not have any concern on the manuscript.